# Duration of immunity following infection with moderately virulent ASFV

**Virginia Friedrichs[1], Paul Deutschmann[1], Kerstin Wernike[1], Tessa Carrau[2], Martin Beer[1], Sandra Blome ![orcid][1]\*, Alexander Schäfer ![orcid][1]**

**1** Friedrich-Loeffler-Institut, Institute of Diagnostic Virology, Greifswald Insel-Riems, Germany,
**2** Department of Virology and Serology, Landesuntersuchungsanstalt für das Gesundheits- und Veterinärwesen Sachsen, Dresden, Germany

\* Sandra.Blome@fli.de

## Abstract

African swine fever virus (ASFV) poses a significant threat to pork production and wild pig populations worldwide. The study assessed the long-term fate and immunity of animals recovering from a moderately virulent ASFV infection, following the principles of a duration of immunity study for live vaccines. Pigs inoculated with the moderately virulent ASFV strain *Estonia14* largely developed mild clinical signs and only transient viremia. Six months after the initial inoculation and once fully recovered, all the animals were challenged with highly virulent ASFV *Armenia08*. Only one of the previously exposed pigs exhibited mild clinical signs, while all control animals showed typical signs of acute, lethal ASF. Moreover, only a subset of pigs inoculated with the ASF strain *Estonia14* displayed temporary detectability of ASFV genomes following challenge infection. Virus isolation corroborated these findings, with low levels of infectious virus in organs of previously inoculated pigs (28 days post challenge). Furthermore, monitoring of ASFV-specific IgM and IgG kinetics enabled the analysis of humoral responses. IgG levels were sustained over the study period and increased slightly upon challenge infection. Lastly, plasma analysis revealed elevated complement factor C3a levels post inoculation and challenge in the recovered pigs, directly correlating with challenge virus presence. In contrast, both C3a and C5a levels were increased in the control group. It could be shown that complement system activation was mediated by the lectin pathway, possibly by interaction of mannose-binding lectins and ASFV particles. This study suggests that protective immunity following recovery can last at least six months. No cases of persistent or chronic disease were observed in convalescent pigs. These findings have implications for both vaccine development and assessment, as well as for disease control strategies including surveillance actions.

**Data availability statement:** All relevant data are within the manuscript and its Supporting Information files.

**Funding:** This work has received funding through the Horizon 2020 ERA-NET Cofund International Coordination of Research on Infectious Animal Diseases (ICRAD), project "ASF-RASH" (VF and SB) and the FLI ASF research network (2821ERA23D to SB). The funders had no role in study design, data collection and analysis, decision to publish, or preparation of the manuscript.

**Competing interests:** The authors have declared that no competing interests exist.

## Author summary

African swine fever virus (ASFV) causes the typically lethal African swine fever (ASF) in pigs, and its spread threatens pig populations worldwide. Vaccines can efficiently protect animals but require detailed understanding of the disease and responses in affected animals. For example, little is known about how long recovered animals are protected from lethal infection. Therefore, we infected pigs with the well-characterized moderately virulent ASFV strain *Estonia14*, which most animals survived with only minor clinical signs. Six months later, we infected the same pigs together with a group of non-immune animals with the highly virulent ASFV strain *Armenia08*. Remarkably, all pigs that had previously recovered from moderately virulent ASFV survived with minor or no clinical signs at all. In contrast, all non-immune pigs died of ASF. Further analyses, like low viral loads and continually high antibody levels in survivors, underscore effective protection. We also show that a part of the innate immune system, the complement system, plays a role during these responses. These results demonstrate that recovery from infection with moderately virulent ASFV can induce robust protection for at least six months.

## 1. Introduction

African swine fever (ASF) has taken a heavy toll on wild pig populations and global pig production driven by factors such as international trade and wildlife movements. The panzootic dimensions and ASF-related mass mortalities and trade restrictions have led to widespread economic losses in associated sectors, e.g., in Europe [1], Asia [2,3], and the Caribbean [4]. In many affected areas, the etiological agent, ASF virus (ASFV), has also caused devastating outbreaks among wild boar populations. These are often the source of spillover infections in domestic pigs raised on farms.

Current knowledge suggests that animals surviving ASFV infection can develop varying degrees of long-term immunity, ranging from partial to full protection. The induction and degree of immunity depends on factors such as strain specific virulence and immune evasion strategies (influencing the number of convalescent animals), as well as host genetics and environmental conditions. In the context of virulence and immune evasion strategies, it has been shown that infection with the naturally attenuated and moderately virulent genotype II ASFV strain *Estonia14* causes mortality rates of 10–20% in domestic pigs, but up to 100% in wild boar [5], whereas infection with a highly virulent strain such as *Armenia08* (genotype II) results in mostly 100% lethality under experimental conditions. Infection of domestic pigs with highly virulent, moderately virulent, and attenuated ASFV strains resulted in varying disease severities; ranging from fatal to transient and mild [6]. A possible reason for such observations is significant differences in immune modulation capabilities between the strains: ASFV was shown to interfere with the host immune response at various levels, such as interferon modulation, inflammation, apoptosis, antigen presentation, and cellular

immunity [7]. Although ASF-recovered pigs showed no evident clinical signs of disease anymore, mild and nonspecific lesions in several organs (e.g., lung and heart) were reported [8]. However, although long-term effects such as scarred and fibrinous lesions in recovered pigs have been described, no evidence of persistent ASFV infection has been found in recent animal trials with strains of genotype I and II [5,9].

Comprehensive knowledge about this duration of immunity against ASFV is key for disease management and control efforts, particularly in enzootic regions where recurrent outbreaks pose significant economic and animal health burdens. Additionally, understanding the kinetics of immunity onset, maintenance, and robustness is essential for vaccination strategies aimed at conferring long-lasting protection against ASFV. This is of particular importance, since continuous research efforts are made towards developing an effective oral vaccine against ASFV. Once developed and licensed for field use, large-scale vaccination campaigns would benefit from comprehensive knowledge about the duration of immunity induced by attenuated field isolates or modified live vaccine strains. Highly virulent isolates like ASFV *Armenia08* will result in lethality rates close to 100% in both domestic pigs and wild boar and, therefore, will not elicit cellular or humoral immunity. However, detection of seropositive wild boar in Germany indicates that although pathogenesis of German ASFV isolates is similar to highly virulent ASFV *Armenia08* [10], wild boar can survive and mount humoral immunity. However, the robustness of this immunity has not been assessed yet. Knowing kinetics and robustness of immunity after infection are essential to determine optimal vaccination schedules and booster frequencies to induce and maintain protective immunity within a pig population. Furthermore, data on the duration of immunity after vaccination could aid implementation of dynamic biosecurity measures. Data on immunity could be also implemented into epidemiological modeling and lead to cost-benefit analysis within risk assessment and biosecurity measures. Movement and trade restrictions might be relaxed in areas with high prevalence of immune animals, while tightened or maintained in areas with low prevalences of immune animals. In addition, relying solely on antibody detection to assess the immune status of an animal may be inadequate for biosafety as recent evidence suggests a potential disconnection between antibody levels and protection capacity. In some cases, antibody levels can wane faster than the actual capability to resist infection and *vice versa* [11].

Taking this into account, we wanted to shed light on the duration of immunity against ASFV in pigs following natural infection. We therefore assessed the long-term protective efficacy induced by a prior exposure to the moderately virulent ASFV strain *Estonia14* against a subsequent challenge infection with the highly virulent strain *Armenia08*. Through analysis of immune parameters such as humoral responses and complement activation, as well as clinical outcomes, we defined suitable assays, i.e., immunoglobulin isotype ELISAs and complement activation, to detect major immune checkpoints in the ASFV-specific immunity. Defining these immune characteristics in pigs recovered from infection with the moderately virulent strain ASFV *Estonia14* might also lead to the definition of correlates of protection, thereby facilitating more targeted future vaccine developments. A challenge of the induced immunity after six months was chosen, because such studies are mandatory for licensing dossiers and the duration of immunity induced by vaccine strains of less than six months would not be sufficient. However, since live attenuated vaccine strains are based on common ASFV strains, definition of the duration of immunity after natural infection is essential to put vaccine strain data into a proper context.

## 2. Materials and methods

### 2.1. Ethics statement

This research was conducted in accordance with German animal welfare regulations, including EU Directive 2010/63/EC and institutional guidelines. Approval was received by the State Office for Agriculture, Food Safety and Fishery in Mecklenburg-Western Pomerania (LALFF MV) and is filed under reference number 7221.3-1.1-004/20.

### 2.2. Experimental design

The animal experiment consisted of two groups of 10 male Large White domestic pigs each. The first group of animals arrived at the Friedrich-Loeffler-Institut (FLI), Riems, Germany, at the age of eight weeks and a weight of approximately

20–25 kg. An age-matched group of 10 female Large White domestic pigs entered the study after six months, to serve as naïve control animals. Prior to transfer to the FLI, all animals were tested negative for ASFV by qPCR. All animals were obtained from the FLI breeding facility in Mariensee and given a two-week acclimatization period. The animal experiment was performed in accordance with the latest German animal welfare regulations. It was approved by the competent authority (Landesamt für Landwirtschaft, Lebensmittelsicherheit und Fischerei Mecklenburg-Vorpommern [LALLF M-V]) under the reference 7221.3-1.1-004/20 before the animals were obtained. To enable indubitable identification of each individual, all animals received a unique ear tag: #91, #92, #93, #94, #95, #96, #97, #98, #99, #100 (first group) and #44, #46, #50, #53, #65, #66, #77, #80, #83, #84 (challenge control group). The two groups were held under similar housing conditions, but in different pens to ensure species-appropriate social behavior.

The first part of the study included oro-nasal inoculation of pigs to mimic natural infection routes without vector involvement. The first group ($n = 10$) was inoculated with 2 ml/pig of $10^{4.25}$ hemadsorbing units 50% ($HAU_{50}$) per ml of the attenuated ASFV strain *Estonia14* (genotype II). This virus strain had shown an attenuated phenotype in previous studies [5] and was chosen to allow survival and assessment of the duration of immunity. Back-titrations confirmed a titer of $10^4$ $HAU_{50}$/ml.

After six months, the surviving animals ($n = 9$), together with the control group ($n = 10$), were inoculated oro-nasally with 10 ml/pig of $10^4$ $HAU_{50}$/ml of the highly virulent ASFV strain *Armenia08* (genotype II). Back-titrations confirmed a titer of $10^4$ $HAU_{50}$/ml. The first group (animals inoculated with both viruses) will be referred to as 'EST + ARM' and the challenge control animals as 'ARM'.

After inoculation, blood and serum were collected weekly during the first month, and monthly for the following five months (days 0, 7, 14, 21, 28, 63, 92, 126, and 154 post infection). After challenge, all animals were sampled on days 0, 4, 7, 10, 14, 21, and 28. Throughout the study, animal well-being was assessed daily using a comprehensive clinical scoring (CS) system. This system allows scoring of changes in animal behavior (e.g., occurrence of appetite loss or lethargy) and appearance (occurrence of skin lesions, hematoma, or swellings) linked to disease progression ([12], modified to include a humane endpoint, if an animal reaches four points in any category). Animals were euthanized either upon reaching a clinical score of 10 points (moderate endpoint, [13]) or upon the development of clinical signs that were classified as intolerable. Furthermore, rectal temperatures were recorded daily and ≥ 40.0°C was defined as fever.

## 2.3. Viruses and cells

For oro-nasal infections, porcine spleens from animals infected with either ASFV *Estonia14* (LALLF M-V reference 7221.3-1-071/21, [14]) or ASFV *Armenia08* (LALLF-MV reference 7221.3-1.1-003/20, [15]) were used. The frozen spleen samples were homogenized in a mortar with sterile sea sand (Roth). The suspension was clarified by centrifugation ($450 \times g$, 15 min, 4°C) and titrated on monocytes/macrophages derived from peripheral blood mononuclear cells (PBMCs) to ensure a sufficient ASFV titer to elicit infection in the animals.

For isolation of PBMCs, blood was obtained from healthy donor pigs that are kept at the FLI quarantine facility. Whole EDTA-blood was mixed with Hanks dextran (10% solution) at a ratio of 1:10. After an incubation of 90 min at room temperature, the PBMC-containing supernatant was collected and washed before seeding in Dulbecco's Modified Eagle's Medium (DMEM, supplemented with 10% fetal calf serum and 0.01% Penicillin/Streptomycin, Gibco). The fraction containing red blood cells (RBCs) was collected and diluted in 1X PBS at a ratio of 1:10 to perform hemadsorption tests (HATs). The HATs were used to a) titrate the inoculum prior to use to ensure sufficient ASFV titers for infection and b) back-titrate it afterwards as validation of the inoculum. The PBMC-containing supernatant was mixed with PBS and centrifuged ($350 \times g$, 10 min, 4°C) and PBMCs were seeded into 96-well plates at a density of $3 \times 10^5$ cells/well and incubated at 37°C in presence of 5% $CO_2$ and a humidified atmosphere. After 24 h of incubation, recombinant colony-stimulating factor 2 (CSF2, Kingfisher Biotech) was added at a concentration of 2 ng/ml to initiate differentiation of monocytes. Following a differentiation period of 24 h, the inoculum (either *Estonia14* or *Armenia08*) was added to the cells (100 µl/well) in a 10-fold dilution series. All titrations were carried out in technical triplicates with four wells per dilution. After 24 h, donor-specific RBCs

were added to each well at a ratio of 1:40. Final titers were determined 48 h later and calculated as 50% hemadsorption dose ($HAD_{50}$) as previously described [16].

## 2.4. Sample collection and processing

To monitor kinetics of ASFV genome and ASFV-specific antibodies in pigs upon infection with moderately virulent ASFV *Estonia14,* blood and serum were taken from each animal on each sampling day. After six months, the surviving animals (group: EST+ARM) were challenged with the highly virulent ASFV *Armenia08* along with challenge control animals (group: ARM). Blood and serum were collected on days 0, 4, 7, 10, 14, 21, and 28 after challenge.

Serum and EDTA-blood were collected from the *vena jugularis externalis* using aspiration collection tubes (KABE Labortechnik), while pigs were restrained with a snout sling or rope. Both matrices were directly subjected to nucleic acid extraction and subsequent qPCR. Part of the EDTA-blood was centrifuged to obtain blood plasma for subsequent analyses of soluble complement factors, i.e., C3a and C5a.

During necropsy, either scheduled for all survivors 28 days post challenge (dpc) or at the humane endpoint for all challenge control animals (7–8 dpc), the following organs were obtained: tonsil, lung, gastrohepatic lymph node (ghLN), mandibular lymph node (maLN), spleen, bone marrow (BM), blood, and serum. Around 200 mg of each organ sample was placed in a 2 ml centrifugation tube containing 1 ml of PBS and a 5 mm metal bead, then homogenized at 30 Hz for 3 min using a tissue lyzer (TissueLyser II, Qiagen). Tissue homogenates, as well as blood and serum were subjected to nucleic acid extraction and qPCR to assess ASFV genome loads. To investigate whether infectious virus can be found in organs of survivors and challenge controls, virus isolation was performed with homogenates of tonsil, ghLN, and spleen.

## 2.5. DNA extraction and PCR

For extraction of nucleic acids, tissue homogenates, whole blood and serum samples were processed using the Nucleo-Mag VET Kit (Macherey-Nagel) on a KingFisher 96 Flex platform (Thermo Fisher Scientific), according to manufacturer's instructions. For whole blood, 70 µl were used for nucleic acid extraction, while 100 µl were used of tissue homogenate or serum. A serum sample negative for ASFV-genome was included as extraction control. For qPCR, the virotype ASFV 2.0 PCR kit (Indical) was used. Each qPCR reaction was carried out according to manufacturer's instructions, multiplexing ASFV genome (FAM), porcine *ACTB* as reference gene control (HEX), and a heterologous internal control to ensure inhibition-free performance (Cy5). All qPCR reactions were carried out on a Bio-Rad C1000 thermal cycler, equipped with the CFX96 Real-Time System (Bio-Rad). ASFV genome copy numbers were calculated by utilizing a PCR standard containing a defined amount of extracted ASFV DNA ($8 \times 10^7$ genome copies/ml) in RSB50 buffer (50 ng/µL carrier RNA, 0.05% Tween 20, 0.05% sodium azide in RNase-free water).

Additionally, to verify whether PCR-positive results in challenged survivors are due to an ASFV *Estonia14* reactivation or due to replication of the challenge virus, an *Armenia08*-specific PCR was employed. All PCR-positive results of surviving animals, as well as all control animals were subjected to PCR. The PCR reactions were setup with already extracted DNA as template, GoTaq Flexi DNA Polymerase (Promega), dNTPs (Jena Bioscience), and specifically designed primers. Primers were designed to amplify part of the ASFV-KP177R gene, which is absent in the ASFV *Estonia14* genome, but present in the ASFV *Armenia08* genome [5]. Primer sequences are as follows: *Forward* 5'-3' CCCAAGAGGTGTGTGAAA, *Reverse* 5'-3' GCATGTTTATGATTTCTAGGTAAGG, with an expected amplicon size of 172 bp. All PCR reactions were run with the following protocol: 1. 95°C for 2 min, 2. 95°C 30 s, 3. 60°C 30 s, 4. 72°C for 1 min (step 2–4 40 cycles), and 5. 72°C 5 min. Amplicons were visualized on an 2.5% agarose gel, containing 1:20.000 ethidium bromide and a GeneRuler 50 bb DNA Ladder (Thermo Fisher) for size reference.

## 2.6. Virus Isolation

Detection of infectious virus in tonsil, spleen, and gastrohepatic lymph nodes was carried out via HAT. However, to ensure accurate results for organ samples containing low amounts of infectious virus, a blind passage was carried out before evaluation of viral loads in organs. For blind passage, PBMCs were seeded into a 24-well plate at a density of $1.5 \times 10^6$ cells/well, in presence of 2 ng/ml CSF2. After 24 h, 200 µl of each organ homogenate was added in duplicates and incubated for 72 h. The plates were frozen at -80°C for ≥ 24 h to ensure rupture of the cells, before ASFV-containing supernatant were used in HAT. Each technical replicate ($n=2$) of the blind passage was subjected to HAT in technical replicates ($n=4$). The results were divided into negative (—, all wells negative), weakly-positive (•, up to 4 wells positive), positive (••, 4–8 wells positive), and strongly positive (•••, 4–8 wells positive, high rosette counts).

## 2.7. Serology

To monitor kinetics of ASFV-specific antibodies, all serum samples were subjected to several serological assays. First, all samples were evaluated using two routinely used and accredited ELISA kits: (I) ID Screen African Swine Fever Competition (ID.vet, Grabels, France), detecting antibodies targeting ASFV p32, and (II) Ingezim PPA COMPAC (Gold Standard Diagnostics), detecting p72 targeting antibodies. All assays were performed in accordance with manufacturer's instructions. However, since these assays either only detect porcine IgG or do not distinguish between Ig isotypes at all, an indirect ELISA protocol was additionally employed to not only evaluate whole ASFV-specific immunoglobulins, but to assess IgG and IgM kinetics individually. Therefore, ASFV-antigen was obtained from the European Union Reference Laboratory for ASF (EURL-ASF), CISA-INIA (Spain). Microtiter plates (medium-binding plate, Greiner) were coated with ASFV-antigen in sodium carbonate buffer, as described in the SOP provided by the EURL-ASF (SOP/CISA/ASF/ELISA/1, REV.5, 2021). In brief, all plates were washed after incubation overnight, dried and stored at -20°C until further use. To reduce unspecific binding, all plates were incubated with a blocking solution (1 × PBS supplemented with 5% horse serum) for 3 h at room temperature. All sera were diluted with washing solution (1 × PBS supplemented with 0.05% Tween-20 (Roth)) at a ratio of 1:30 immediately prior to performing the experiment. Reference sera from the German National Reference Laboratory for ASF (NRL-ASF) were included on each plate as positive and negative controls. For normalization purposes, two wells on each plate were not coated with ASFV-antigen and only treated with sodium carbonate coating buffer. After incubation and subsequent washing, the respective, HRP-conjugated secondary antibody was added to each well: for (I) IgG detection, a polyclonal anti-porcine IgG antibody (#A100-205P, Biomol) was added at a dilution of 1:10,000, whereas for (II) IgM detection, a polyclonal anti-porcine IgM antibody (#AAI48, Bio-Rad) was added at a dilution of 1:5,000. ELISA TMB Stabilized Chromogen (ThermoFisher) was added as substrate. The reaction was stopped with ELISA Stop Solution (ThermoFisher) and the signal was acquired with an ELISA reader (TECAN) at a wavelength of 450 nm.

Activated complement factors C3a and C5a were measured using species-specific and commercially available kits according to manufacturer's instructions (MyBioSource, San Diego, USA). Evaluation of C3a and C5a levels was conducted with plasma of the pigs, as plasma samples are more accurate in detecting soluble biomarkers compared to serum samples [17]. For *in vitro* assays, untreated serum from animals of this study taken 0, 7, and 154 dpi was mixed 1:2 with either PBS, 10 mM EDTA to achieve complete complement inhibition, or 1 mM EGTA/2 mM MgCl$_2$ (MgEGTA) to inhibit classical and lectin pathway, and incubated for 30 min at 37 °C. Afterwards, the treated serum samples were mixed with $10^6$ HAD$_{50}$/ml *Armenia08* and incubated for 90 min at 37 °C. The reaction was stopped by addition of 10 mM EDTA and 50 µg/ml Futhan (BD Biosciences) to completely inhibit all complement pathways and other protease activity. Levels of C3a and C5a were investigated by C3a and C5a ELISA as described above. For analysis of complement-dependent infection rates, untreated or heat-inactivated (56 °C, 30 min) plasma samples of six naïve animals were incubated with $10^6$ HAD$_{50}$/ml particles of either ASFV *Estonia14* or ASFV *Armenia08* for 90 min at 37 °C. The reaction was stopped by addition of 10 mM EDTA and 50 µg/ml Futhan (BD Biosciences). Pretreated ASFV particles were used for infection of porcine monocytes (prepared 48 h in advance). Infection rates were assessed 48 h later by indirect immunofluorescence

with anti-ASFV-p72-FITC (polyclonal rabbit serum, in-house) and DAPI counterstain on a Leica Thunder Imager (Leica Microsystems). Using the automated event-counting function of ImageJ, all cells/well were counted (all DAPI-positive events and the ratio of infection was calculated by counting and normalizing infected cells (FITC-positive events) to total cell counts.

### 2.8. Statistics

Statistical analyses and data visualization were performed with GraphPad Prism 10.3.1 for Windows (GraphPad Software Inc.). Sample numbers are indicated in the figure legends. Normality was verified by D'Agostino-Pearson normality test. To assess differences in survival and clinical scores between previously *Estonia14*-infected and control animals, area under the curve (AUC) was calculated for each group with clinically inapparent control times as baseline and only positive peaks considered. Total area and standard error of both AUCs were then analyzed by unpaired *t*-tests with df calculated as the number of data points minus the number of time points. Significant differences in complement activation were analyzed by a mixed model with Holm-Šidák's correction for multiple comparisons. Differences between serum treatment groups were investigated by paired t-test (correction for multiple comparisons by False Discovery Rate). Significant statistical differences are indicated by their *p*-value. Compact letter display shows statistically significant differences between multiple groups. Groups that don't share any letter (e.g., ac and de) have statistically significant differences between them ($p < 0.05$), while groups with one or more shared letters (e.g., ac and cd) have no statistically significant difference.

## 3. Results

### 3.1. Course of ASFV Armenia08 infection in pigs previously infected with ASFV Estonia14

After inoculation with ASFV *Estonia14*, all 10 animals developed fever and first clinical signs of disease 4–5 dpi (Fig 1). The body temperature was elevated in all animals 5 dpi ranging from 39.5°C to 41.8°C until 10 dpi (Fig 1A). Clinical signs of disease were mild in all animals in the first week of infection, where lethargy, loss of appetite, and reduced liveliness could be observed (Fig 1B). One individual (#94) succumbed to infection and was found dead in the pen at 9 dpi. All other animals fully recovered until 14 dpi, with no elevated body temperature or clinical signs. The ASFV genome copy numbers in *Estonia14*-infected animals were comparable in all animals in the first 4 weeks after inoculation (Fig 1C). The detected ASFV genome copy numbers were highest 7 dpi with $4.3 \times 10^8$/ml blood and decreased slowly to $4 \times 10^7$/ml blood 28 dpi. During the next months, the animals were sampled once a month. By month 4, only one animal was still ASFV-positive in blood ($1.6 \times 10^4$/ml blood) and all animals were negative five months after infection.

After an observation period of six months, all remaining pigs, along with a control group, were inoculated with ASFV *Armenia08* to assess the robustness of the ASFV-specific immunity. Only one individual (#93) in the EST+ARM group developed fever after challenge infection, however, the fever did not persist (Fig 2A). The clinical score of this animal on the days of fever included scores for gait and bearing, possibly resulting from bacterial infection between the claws, not ASFV-specific manifestations. In contrast, all animals of the ARM group developed high fever, ranging from 40°C 5 dpi to 41.8°C 8 dpi (Fig 2B). Onset of fever in the control animals was accompanied by a disease manifestation typical for ASFV infection: severe lethargy, no appetite, dehydration, pain in extremities with difficulties standing/walking, and respiratory distress. All control animals reached the humane endpoint 7–8 dpi. ASFV genome copy numbers in the ARM group peaked at around $6.7 \times 10^9$/ml blood 7 dpc. Among the EST+ARM group, five animals were ASFV-positive at least once (Fig 2C), while the remaining animals remained negative during the challenge phase (all positive for ASFV *Armenia08* not ASFV *Estonia14*, as shown in a differentiating PCR (S1 Fig)). However, ASFV genome copy numbers were considerably lower than in control animals, peaking at $2.3 \times 10^6$/ml blood.

Gross pathological findings in the ARM group and #94 of the EST+ARM group comprised of typical manifestations during ASFV infection: hemorrhages in several organs (e.g., petechiae in kidneys), ascites, and pericardial effusion (see S1 File). In contrast, gross pathology in the EST+ARM group rendered no abnormalities, except scarred and fibrinous

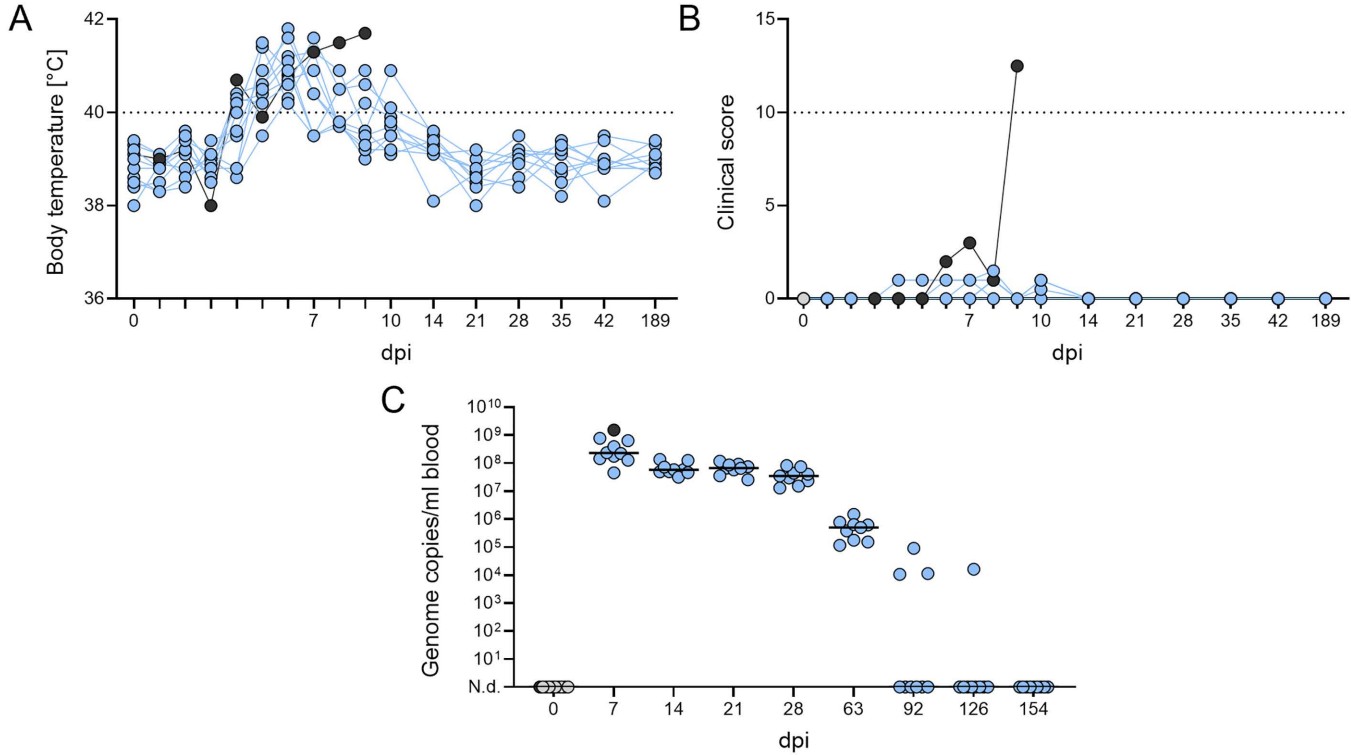

**Fig 1. Clinical course and viral load in blood after inoculation with the moderately virulent ASFV strain *Estonia14*. (A)** Body temperature, **(B)** clinical score, and **(C)** ASFV genome loads in blood of pigs (*n* = 10) upon inoculation with ASFV *Estonia14*. Black symbols show one individual (#94) that developed severe clinical symptoms and succumbed to infection 9 dpi. The dotted lines indicate thresholds for fever (A) and the humane endpoint **(B)**. Lines in panel C represent medians, symbols in all panels represent individual animals.

lesions in and around the pericardium of #93, #97, #99, and #100, indicating a previous pericardial effusion or bacterial colonization (see S1 File). The ASFV genome loads were assessed in several organs (Fig 3). In the ARM group, all investigated tissues exhibited high genome copy numbers in all animals. In contrast, only some animals of the EST + ARM group were weakly positive in organs at the time of euthanasia. Of note, ASFV genome copies in those animals originated from the challenge virus, as an ASFV *Estonia14*-specific qPCR was negative for blood and organ samples.

Subsequently, tonsil, gastrohepatic LN, and spleen samples were subjected to virus isolation (Table 1). High amounts of infectious virus particles were found in all animals of the ARM group, as well as in animal #94 of the EST + ARM group. Moderate amounts of infectious particles were found in tonsil and ghLN samples of #93, #97, #99, and #100, while spleen samples were weakly positive.

### 3.2. Kinetics of IgM and IgG in serum of pigs recovered from ASFV infection

To assess the humoral response in after *Estonia14* inoculation and *Armenia08* challenge infection, commercially available competitive ELISAs and newly established ELISAs based on the EURL-ASF ELISA to investigate ASFV-specific IgM and IgG kinetics were used (Fig 4). The competitive ELISAs targeting the early ASFV-p32 protein (Fig 4A) and the late ASFV-p72 protein (Fig 4C) showed induction of humoral immunity at early time points after *Estonia14* inoculation, with the first seroconverted animals 7 dpi for ASFV-p32 and 14 dpi for ASFV-p72. Anti-ASFV-p32 antibodies remained stable at high levels over the whole study period. Anti-ASFV-p72 antibodies reached their maximum 63 dpi and also remained stable throughout the trial. Animal #94, that succumbed to *Estonia14* inoculation 9 dpi, did not seroconvert. Antibodies

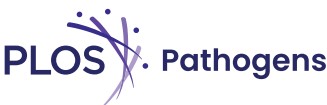

**Fig 2. Clinical course and viral load in blood after challenge infection with the highly virulent ASFV strain *Armenia08*. (A)** Body temperature and **(B)** clinical score of previously *Estonia14*-inoculated (*n* = 9, blue) and naïve pigs (*n* = 10, red) after challenge with ASFV *Armenia08*. **(C)** ASFV genome copy numbers in blood of challenged animals. Genome copy numbers were calculated per 1 ml blood. Mind the different scales in (C) for previously inoculated and naïve animals. Numbers indicate ear tags. The dotted lines indicate thresholds for fever (A) and the humane endpoint **(B)**. N.d., Not detected.

against ASFV-p32 and -p72 remained stable after challenge infection in previously inoculated animals. Naïve animals did not seroconvert after challenge infection (Fig 4B, 4D). However, these commercially available ELISAs cannot differentiate between Ig isotypes and fail to detect minor changes in Ig levels. Therefore, we also established ELISAs employing whole virus lysate and Ig isotype detection to assess the kinetics of ASFV-specific IgM and IgG.

We detected high levels of ASFV-specific IgM 7 and 14 dpi after inoculation with *Estonia14* (Fig 4E). These declined over the course of the study but remained above threshold in most animals. Upon challenge with *Armenia08*, levels of IgM increased again in all animals but were overall lower than after *Estonia14* inoculation and declined rapidly. First detection of ASFV-specific IgG after inoculation with *Estonia14* was possible in most animals 14 dpi (Fig 4G). IgG titers increased

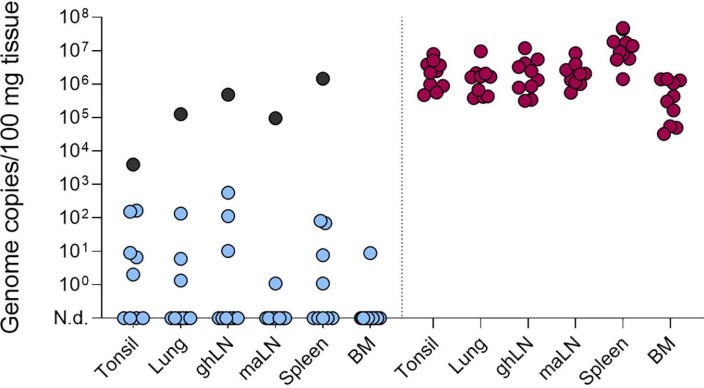

**Fig 3. Viral load at necropsy in various tissues after challenge infection with the highly virulent ASFV strain *Armenia08*.** ASFV genome copy numbers after challenge with ASFV *Armenia08* in organs of pigs previously infected with *Estonia14* (*n* = 9, blue, 28 dpc) or naïve animals (*n* = 10, red, 7-8 dpc). Genome copy numbers were calculated per 100 mg tissue. Symbols represent individual values. Black symbols show animal #94, that died 9 dpi. ghLN, gastrohepatic lymph node; maLN, mandibular lymph node; BM, bone marrow, N.d., Not detected.

**Table 1. Virus isolation results in organs of pigs previously infected with *Estonia14* (EST + ARM) or naïve animals (ARM).**

| EST + ARM | | | | ARM | | | |
|---|---|---|---|---|---|---|---|
| | Tonsil | ghLN | Spleen | | Tonsil | ghLN | Spleen |
| #91 | — | — | — | #44 | •• | ••• | ••• |
| #92 | — | — | — | #46 | •• | ••• | ••• |
| #93 | •• | •• | •• | #50 | •• | ••• | ••• |
| **#94** | ••• | ••• | ••• | #54 | ••• | ••• | ••• |
| #95 | — | — | — | #65 | ••• | ••• | •• |
| #96 | — | — | — | #66 | ••• | ••• | ••• |
| #97 | •• | •• | • | #77 | ••• | ••• | •• |
| #98 | — | — | — | #80 | ••• | ••• | ••• |
| #99 | •• | • | • | #83 | •• | ••• | ••• |
| #100 | •• | •• | • | #84 | ••• | ••• | •• |

— = negative (0/8); • = weakly positive (1-4/8); •• = positive (5-8/8); ••• = highly positive (8/8 with high rosette count), ghLN, gastrohepatic lymph node

**Bold** = animal succumbed to ASFV *Estonia14* infection 9 dpi

over time and remained relatively stable between months 2 and 6 after inoculation. Upon challenge, IgG levels moderately increased 14 and 21 dpc, but were comparable to 0 dpc at the end of trial (28 dpc). Of note, previously *Estonia14*-inoculated animals that were positive for infectious virus after challenge infection did not show different antibody levels then animals that remained negative (Fig 4E, 4G). Animal #94 remained negative in both ELISAs. Naïve animals did not seroconvert after challenge infection (Fig 4F, 4H).

While these results rendered comparable results to commercially available competitive ELISAs, the adapted isotype-specific ELISA enabled the analysis of the humoral response with greater detail.

### 3.3. Complement activation upon infection with moderately and highly virulent ASFV

To further evaluate soluble inflammation factors, plasma was collected on all sampling days and subjected to species-specific investigation of complement activation by detection of complement components 3 (C3a) and 5 (C5a) in all animals (Fig 5). During *Estonia14* infection, C3a levels significantly increased in EST + ARM animals 14 dpi (Fig 5A). After

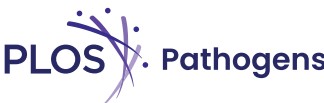

**Fig 4. Humoral responses in pigs after inoculation with the moderately virulent ASFV *Estonia14* and challenge infection with the highly virulent ASFV *Armenia08*.** Commercially available competitive ELISAs were used to assess kinetics of antibodies against **(A, B)** ASFV-p32 and **(C, D)** ASFV-p72 in sera of (A, C) pigs inoculated with ASFV *Estonia14* and then challenged with ASFV *Armenia08* (blue, $n = 10$) or (B, D) naïve pigs (red, $n = 10$). Grey areas between dotted lines indicate assay threshold, values above were considered positive. ASFV-specific antibodies and their isotypes

were assessed by ELISA. Kinetics of **(E, F)** IgM and **(G, H)** IgG in sera of (E, G) pigs after inoculation with ASFV *Estonia14* and challenge with ASFV *Armenia08* (blue, *n* = 10) and (F, H) naïve pigs after challenge with ASFV *Armenia08* (red, *n* = 10). Dotted lines indicate cut-off values. Animal #94 that succumbed to *Estonia14* inoculation is shown in black. Animals that were still positive in qPCR at the end of the trial (28 days after challenge) are shown as bright red squares. Lines represent means, symbols individual values. Mixed-effects analysis (EST+ARM) or repeated-measures ANOVA (ARM) with Holm-Šidák's correction for multiple comparisons. Compact letter display shows statistically significant differences, with different letters indicating *p* < 0.05.

**Fig 5. Activation of the complement system after ASFV infection.** Detection of activated complement components **(A)** C3a and **(B)** C5a in plasma after inoculation with ASFV *Estonia14* and challenge with ASFV *Armenia08* (*n* = 9, blue), or after infection in control animals (dark red, *n* = 10). The respective reference days (day 0 before infection/challenge) are displayed in grey. Lines represent means, symbols show individual values. Bright red squares among EST+ARM animals indicate individuals where live virus was isolated 28 dpc. Two-way-ANOVA with Holm-Šidák's correction for multiple comparisons between the timepoints and the respective controls. **(C)** Correlation analysis with data from 28 dpc in EST+ARM animals (left and middle) and EST+ARM vs. ARM animals 7 dpc (right). N.d., not detected.

*Armenia08* challenge, C3a levels in EST+ARM animals increased slightly but only reached significance at 28 dpc. C5a levels remained unaffected during both infections in the EST+ARM group (Fig 5B). In ARM animals, levels of C3a significantly increased during *Armenia08* infection, as early as 4 dpc. Contrary to EST+ARM animals, the levels of C5a were significantly increased in ARM animals 7 dpc.

Of note, the four animals with the highest C3a levels among EST+ARM animals after challenge infection were animals which still contained infectious virus in organs at the end of the trial (#93, #97, #99, #100). Cq-values and genome copy numbers in EST+ARM animals were positively correlated with C3a levels 28 dpc ($R^2 = 0.8203$, $p = 0.0008$ and $R^2 = 0.5339$, $p = 0.0253$; Fig 5C). Likewise, C5a levels correlated with body temperature 7 dpc ($R^2 = 0.7149$, $p < 0.0001$, Fig 5C, right). In contrast, IgG titers were not predictive of C3a levels and Cq values did not correlate with IgG levels (S2 Fig).

Additionally, we preliminarily investigated the mechanism behind complement activation in additional *in vitro* assays. Serum of pigs at different stages after *Estonia14* infection was incubated directly with ASFV *Armenia08*. The stages were chosen based on the presence of serum antibodies to assess the classical complement pathway: naïve serum (0 dpi) contained no antibodies and late serum (154 dpi) contained high levels of ASFV-specific IgG (Fig 4A, 4B). In order to identify the complement pathway that led to activation, we further added either PBS (control, no inhibition), EDTA (complete complement inhibition), or EGTA/MgCl$_2$ (MgEGTA, specific inhibition of classical and lectin pathway) to the serum. We found a significant decrease of C3a and C5a levels after incubation with EDTA and MgEGTA (Fig 6A, 6B). Subsequently, we explored whether decoration of ASFV particles by complement factors would increase viral uptake and infection rates. Viral particles of *Estonia14* or *Armenia08* were incubated with untreated or heat-inactivated plasma from naïve animals and used for *in vitro* infection of porcine monocytes for 48 h. Infections rates were assessed by indirect immunofluorescence. We found a significant increase when *Estonia14* particles were incubated with heat-inactivated plasma, but no significant difference for *Armenia08* (Figs 6C, S3).

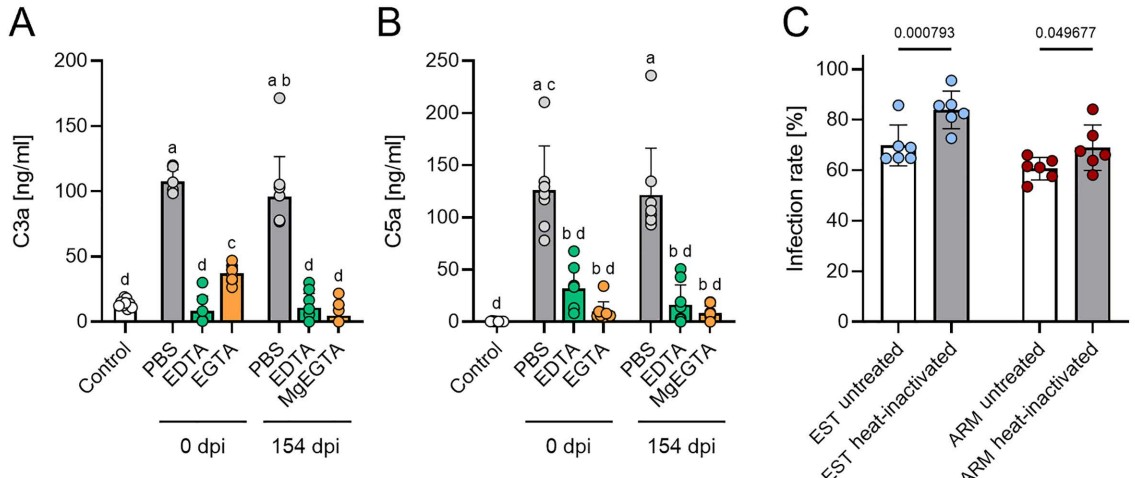

**Fig 6. *In vitro* investigation of complement system responses against ASFV.** Detection of activated complement components **(A)** C3a and **(B)** C5a *in vitro*. Untreated serum of the indicated times after *Estonia14* inoculation (plasma of 154 dpi contains ASFV-specific IgG) was mixed with PBS (grey), EDTA (green), or EGTA/MgCl$_2$ (MgEGTA, orange) and then incubated with $10^6$ HAD$_{50}$/ml ASFV *Armenia08* ($n = 7-9$). Complement activation was assessed by C3a- and C5a-specific ELISAs. **(C)** Infection rates of monocytes 48 h after infection with *Estonia14* (blue) or *Armenia08* (red) particles that were pre-incubated with either untreated (white bars) or heat-inactivated (grey bars) plasma from naïve animals ($n = 6$, in triplicates) for 90 min at 37 °C. One-way-ANOVA with Holm-Šidák's correction for multiple comparisons. Compact letter display shows statistically significant differences, with different letters indicating $p < 0.05$. EST, ASFV *Estonia14*; ARM, ASFV *Armenia08*.

## 4. Discussion

Long-term studies on potential protective immunity against ASFV infections provide important data for duration of immunity, improved vaccine designs and potential management option for this notifiable disease. However, these studies are challenging, due to their high demands to animal husbandry within high-containment research facilities. In the present study, inoculation with the moderately virulent ASFV Estonia14 induced protective immunity against the highly virulent ASFV Armenia08 for at least six months in domestic pigs. Interestingly, although all animals were protected from clinically apparent disease, viral genomes were still detected and live virus was isolated from some animals after challenge infection, although titers of recovered ASFV were not assessed.

Previous studies with various ASFV strains demonstrated varying degrees of protection against lethal challenge infection. For example, inoculation with the naturally attenuated ASFV OURT88/3 did not convey protective immunity against highly virulent Benin 97/1 130 days after inoculation [18], contrasting the robust and protective immunity for at least six months (189 days) observed in our study. These observations possibly indicate the development of strain-specific immunity, varying in duration and strength. Such differences may result from strain-specific variations in the immunogenicity of ASFV antigens and capsid-specific antibody responses elicited by different ASFV strains, ultimately altering humoral immune responses [19,20]. Moreover, both ASFV OURT88/3 and Benin 97/1 belong to genotype I, while Estonia14 and Armenia08 belong to genotype II. In addition, the animals in the aforementioned study were intramuscularly infected, while the animals in this study were oro-nasally inoculated and challenged. Our approach mirrors natural infections and does not circumvent natural barriers, probably inducing different immune responses than intramuscular infections, which might explain the differences in the observed protection.

In a recent report using the same ASFV strains as in this study, Radulovic *et al*. showed somewhat contrasting results [21]: They infected conventionally-held and specific-pathogen free (SPF) domestic pigs oro-nasally with ASFV *Estonia14*. Similar to our study, all pigs survived this initial infection with minor clinical signs. However, when subsequently challenged oro-nasally with ASFV *Armenia08* around five months later, all farm pigs developed severe clinical signs, with a lethality rate of 60% in this group (14 dpc). All SPF pigs remained clinically healthy. Of note, ASFV was found by PCR after challenge in blood of all animals regardless of genetic background but at higher levels in farm pigs. While these disparate outcomes despite a similar experimental approach highlight the variability in ASFV infections, there are also some differences that might explain the outcomes. For one, the animals in our study were slightly younger when initially infected (9–10 vs. 12 weeks) but had approximately a month more between initial infection and challenge (189 vs. 164 days). Moreover, while the infectious doses are not directly comparable, they were considerably lower during the initial infection ($2 \times 10^{4.25}$ $HAU_{50}$ vs. $\sim 5 \times 10^9$ ASFV genome equivalents) and the challenge ($10^5$ $HAU_{50}$ vs $10^6$ $TCID_{50}$) in this study. Interestingly, intranasal infection with low doses ($10^3$-$10^4$ $TCID_{50}$) of ASFV *OURT88/3* resulted in complete protection while animals immunized with higher doses ($10^5$ $TCID_{50}$) were not all protected [22]. This might demonstrate what has been suggested as the 'Goldilocks model', which states that (among other factors) the amount of antigen has to be just right to induce long-lasting responses with superior recall and effector capabilities [23]. Likewise, maturation processes can take months, especially if there is still antigen present [23–25]. A longer differentiation period before challenge might therefore even be beneficial. Likewise, a high challenge dose might cause breakthrough infection, as has been observed for multiple other vaccines [26–28], that do not necessarily mirror doses of challenge virus under field conditions.

It has been demonstrated that such differences in antibody responses occur naturally between breeds of domestic pigs and even between individuals within a breed [20]. These conclusions were further validated by observing no such differences in the capsid-specific antibody response in Babraham pigs after inoculation with the moderately virulent ASFV strain *OURT88/3* [29]. Babraham pigs are inbred and express homozygous SLA molecules, excluding the influence of individual genetic and immunological factors [30]. However, although some novel antigens were introduced by the *Armenia08* strain in the 5'-region of the viral genome with the challenge infection ([5], evidenced by increasing IgM levels in

EST+ARM pigs after challenge), strain-specific immunity conveyed by ASFV *Estonia14* was sufficient to protect from a different highly virulent strain, ASFV *Armenia08*.

In addition to humoral responses, cellular immunity plays a protective role against AFSV infection. While this was not a focus of our study, previous data indicates that protection against fatal challenge infections relies heavily on T cell responses [11,31]. In a recent preprint, Lotonin *et al.* used a systems immunology approach to describe correlates of protection after primary infection with *Estonia14* and subsequent challenge with *Armenia08*. For example, they describe induction of ASFV-specific T helper cells prior to challenge and strong recall responses of helper and cytotoxic memory T cells after challenge as predictors of survival [32]. It can, therefore, be assumed that *Estonia14* inoculation in this study also resulted in a robust and protective T cell response.

Furthermore, host genetics and the environmental status of pigs can significantly impact disease outcome. For example, inoculation of domestic pigs and wild boar with ASFV *Estonia14* under standardized conditions resulted in no lethality in domestic pigs and 60% lethality in wild boar [5]. These differences in disease manifestation and outcome may be due to distinct response patterns of CD8+cytotoxic T cells in domestic pigs and wild boar [33]. The environmental conditions in housing facilities of pigs also influences the disease outcome: upon inoculation with ASFV *Estonia14*, SPF pigs had reduced capacity to control early virus replication, but presented overall milder disease with full protection and recovery compared to conventionally housed pigs [34]. In line with this, the data of our trial indicates that immune pigs rarely develop clinical signs upon challenge infection. In addition to the finding of ASFV genomes in the blood and the spleen of some immune animals, infectious ASFV particles were also found. Whether the amount of challenge virus shed by clinically inapparent pigs would suffice to infect naïve penmates remains to be determined. The possible impact of such asymptomatic carriers in wild boar populations might be detrimental and needs to be addressed in future risk assessment discussions. A possible limitation of our study is the sex difference of the groups: EST+ARM, composed of male pigs, and ARM, composed of female pigs. However, while immunological treatments like vaccinations are known to have sex-specific differences [35], ASFV has never been shown to cause different pathologies or other sex-related alterations [19,36–38].

A more comprehensive characterization of antiviral responses in ASFV-infected pigs is pivotal for an improved understanding of ASF pathogenicity and host responses. Detection of disease correlates in easily accessible matrices like serum could increase our understanding of ongoing infections. We found increased levels of complement factors C3a but not C5a in ASFV *Estonia14*-infected animals. Challenge infection with *Armenia08* of recovered animals resulted in similar responses, but only in animals that were found to shed live challenge virus. In contrast, infection with *Armenia08* in naïve pigs resulted in a concurrent increase of both C3a and C5a levels.

The complement system can be activated by three distinct pathways: the classical pathway, using antibody complexes on the target membrane, the lectin pathway, using mannose-binding lectin (MBL) and ficolins detecting specific membrane-bound carbohydrates, and the alternative pathway, which is activated by spontaneous hydrolysis of complement factor C3. The complement system also reacts to viral infections for which many viruses evolved escape strategies [39]. To further characterize the involvement of complement during ASFV infection, we conducted *in vitro* experiments by specifically inhibiting or excluding all or some complement pathways: EDTA, inhibiting all complement pathways, MgEGTA, specifically inhibiting the classical and lectin pathways [40], and serum containing no ASFV-specific antibodies, excluding the classical pathway. We found a significant increase of both C3a and C5a when serum of naïve and recovered animals was incubated with ASFV. This was blocked by MgEGTA, excluding the alternative pathway. Samples taken 0 dpi showed complement induction in the absence of ASFV-specific antibodies, indicating that the classical pathway is not essential in this context. This leaves the lectin pathway as the inducer of complement responses during ASFV infections. While MBL and ficolins are often thought to specifically bind to bacterial carbohydrates [41,42], they have been shown to detect a variety of viruses, including human immunodeficiency virus (HIV [43]), hepatitis B virus (HBV [44]), hepatitis C virus (HCV [45]), SARS-CoV-2 [46], human cytomegalovirus/ human

herpesvirus-5 (CMV/HHV-5 [47]), Ebola virus (EBOV [48]), adenovirus 2 and 5 [49], Chandipura virus (CHPV [50]), Dengue virus [51], and West Nile virus (WNV [52]), in part by binding directly to viral surface proteins. Interactions of the host complement system with a number of ASFV proteins have been shown in an interactome study with the virulent ASFV HLJ/18 strain before [53]. The effects of complement activation during infection might be anti- as well as pro-viral [39,54], but are usually pro-inflammatory [39]. Increased C3a levels are also indicative of heightened levels of C3b, responsible for opsonization of pathogens and might thus contribute to increased uptake of opsonized particles in myeloid cells. However, heat-inactivation of the complement system led to increased infection rates of macrophages of *Estonia14* but not *Armenia08* particles, indicating that at least some viral particles might be neutralized by the activated complement system, which might explain some of the observed attenuation of *Estonia14*. These results are preliminary due to the lack of specific reagents and thus, future mechanistic studies are necessary to verify the suspected complement activation by the lectin pathway during ASFV infection.

Independent of the immunological processes behind the observed variations, understanding the robustness and duration of immunity after infection with ASFV is essential for advancing disease management, control strategies, and risk assessment in both enzootic and non-enzootic regions. The disease has a major economic impact on the pork industry due to high fatality rates, indiscriminate culling, and trade restrictions installed to prevent viral spread. If immunity after ASFV infection or vaccination proves to be robust and long-lasting, it could fundamentally reshape control measures. For instance, evidence of a durable immunity could support more targeted depopulation policies. A satisfactory number of successfully vaccinated animals could lead to a reduction of the need for indiscriminate culling, ultimately minimizing economic losses. On the other hand, if immunity even after vaccination is incomplete or short-lived, stringent biosecurity measures like rapid diagnostics, and whole farm cullings remain a necessity. Furthermore, a reduced duration of immunity might also influence future vaccination concepts like booster intervals.

In Europe, studies for identifying the duration of immunity after vaccination are mandatory to be included in a dossier for licensing a vaccine candidate. Once the data is collected and a vaccine is fully licensed, knowing how long the induced immunity will convey protection is crucial for the development of vaccination strategies. Given that the duration of immunity is at least six months, but varies between strains (and possibly ASFV vaccine strains as well), large-scale vaccination campaigns for wild pigs can be optimized and refined, especially the regimen for booster vaccinations or the need for multi-component vaccines targeting different viral mechanisms. It is important to understand whether natural infection provides lasting immunity in order to guide the design of vaccines, which mimic such protection. The data about the duration of immunity after inoculation with an attenuated ASFV field strain or a vaccine strain can be implemented into epidemiological modeling algorithms to predict how many boosters at what time frames will be necessary to sufficiently immunize a wild population and hamper viral spread. Such modeling could predict how and if ASFV sustains itself after populations were immunized. Additionally, if animals lose immunity over time, populations may become susceptible to reinfection, necessitating the reevaluation of biosecurity timelines and quarantine durations, particularly in high-risk zones.

## 5. Conclusions

Our study investigated the duration and robustness of immunity induced by a moderately virulent ASFV strain, *Estonia14*, against a related highly virulent strain, *Armenia08*. Pigs that survived an infection with *Estonia14* exhibited no clinical signs upon challenge infection with *Armenia08*, while control animals developed acute disease. However, some *Estonia14*-inoculated pigs showed limited viral presence post-challenge. Stable IgG levels post-inoculation and a moderate increase post-challenge further indicate prolonged maintenance of IgG levels and reactivation of humoral immune responses upon challenge. Interestingly, elevated complement factor C3a levels correlated with challenge virus presence in *Estonia14*-inoculated pigs, suggesting involvement of the complement system in ASFV manifestation. Conversely, increased C3a and C5a levels in control animals may indicate contribution of the complement system to ASF

pathogenicity. These findings demonstrate that prior infection with a moderately virulent ASFV strain elicits robust protection against infection with a highly virulent strain for at least six months.

## Supporting information

**S1 Fig. ASFV strain detection after challenge.** PCR gel bands of ASFV *Armenia08*-specific PCRs in blood samples from (A) EST+ARM and (B) ARM animals of the indicated times after challenge infection. EST+ARM animals that remained negative after challenge were not further investigated. Stocks of ASFV *Estonia14* and ASFV *Armenia08* were used as controls. Red numbers indicate base pair sizes of the 50 bp marker.
(TIF)

**S2 Fig. Correlation analyses that did not reach statistical significance.** Correlation analyses of (A) ASFV-specific IgG and C3a in EST+ARM animals, (B) ASFV Cq values and ASFV-specific IgG in EST+ARM animals, (C) clinical score and C5a levels and (D) body temperature and C3a levels in EST+ARM and ARM animals. EST+ARM animals are shown in blue, ARM animals in dark red. Bright red squares among EST+ARM animals indicate individuals where live virus was isolated 28 dpi.
(TIF)

**S3 Fig. Representative images of the immunofluorescence analysis.** Macrophages were *in vitro* infected with (A) ASFV *Estonia14* (EST) and (B) ASFV *Armenia08* (ARM) pre-incubated either with untreated (upper rows) or heat-inactivated (bottom rows) porcine serum. Infection rates were assessed by detection of ASFV-p72 after 48 h.
(TIF)

**S1 File. Source data.**
(XLSX)

## Acknowledgments

We would like to thank all animal caretakers: Matthias Jahn, Domenique Lux, Patrice Mary and Steffen Brenz for excellent animal husbandry and assistance during sampling procedures. Additionally, we would like to thank Christian Loth and Ralf Redmer for preparing and assisting all necropsies. For excellent technical assistance during sample processing and subsequent experiments, we thank Ulrike Kleinert.

## Author contributions

**Conceptualization:** Virginia Friedrichs, Martin Beer, Sandra Blome, Alexander Schäfer.

**Data curation:** Virginia Friedrichs, Alexander Schäfer.

**Formal analysis:** Virginia Friedrichs, Sandra Blome, Alexander Schäfer.

**Funding acquisition:** Sandra Blome.

**Investigation:** Virginia Friedrichs, Paul Deutschmann, Kerstin Wernike, Tessa Carrau, Sandra Blome, Alexander Schäfer.

**Methodology:** Virginia Friedrichs, Paul Deutschmann, Kerstin Wernike, Tessa Carrau, Sandra Blome, Alexander Schäfer.

**Project administration:** Sandra Blome.

**Supervision:** Martin Beer, Sandra Blome, Alexander Schäfer.

**Visualization:** Virginia Friedrichs.

**Writing – original draft:** Virginia Friedrichs, Sandra Blome, Alexander Schäfer.

**Writing – review & editing:** Virginia Friedrichs, Kerstin Wernike, Martin Beer, Sandra Blome, Alexander Schäfer.

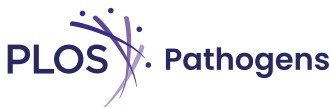

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
