## [Decision Letter · Decision Letter 0]

6 Jan 2026

PPATHOGENS-D-25-02903

Duration of immunity following infection with moderately virulent ASFV

PLOS Pathogens

Dear Dr. Blome,

Thank you for submitting your manuscript to PLOS Pathogens. After careful consideration, we feel that it has merit but does not fully meet PLOS Pathogens's publication criteria as it currently stands. Therefore, we invite you to submit a revised version of the manuscript that addresses the points raised during the review process.

We look forward to receiving your revised manuscript.

Kind regards,

Linda Kathleen Dixon

Academic Editor

PLOS Pathogens

Donna Neumann

Section Editor

PLOS Pathogens

Sumita Bhaduri-McIntosh

Editor-in-Chief

PLOS Pathogens

orcid.org/0000-0003-2946-9497

Michael Malim

Editor-in-Chief

PLOS Pathogens

orcid.org/0000-0002-7699-2064

**Additional Editor Comments:**

This is a resubmission of a previously submitted manuscript by the same authors (Friedrichs et al.,PPATHOGENS-D-25-01346 ) to PLOS Pathogens. The authors have provided a cover letter and a response to the previous reviews. A similar manuscript (Radulovic et al., 2025 DOI: 10.3389/fvets.2025.1553310) was published soon after the first submission by Friedrichs et al..

As requested by the reviewers the authors have included in the discussion section (starting on Line 444) a comparison of their experiments with those of Radulovic at al., 2015. Both studies use similar ASFV strains to study responses in pigs following infections with a moderately virulent ASFV strain (Estonia14) and challenge after a period of 6 months with a virulent strain (Armenia 08). However, some details of the experiments and the outcomes of the infections and challenge differed. For example, Radulovic et al., compared responses in farm pigs and SPF pigs. Friedrichs et al., included additional analyses, including analysis of complement activation, that were not included in the Radulovic et al., 2015 article. Few publications have studied immune mechanisms that may correlate with induction of longer-term protection by attenuated ASFV. This is a critical knowledge gap for the development of safe and effective vaccines. Thus, this article contains novel data of significant interest to the field.

We agree with the reviewers that some minor points remain to be addressed as presented in their reviews and summarised below.

Reviewer 1.

States that: “The authors have addressed the 3 major recommendations and all the minor comments I previously made.

As previously noted, the study provides new and important insight into the duration of immunity against ASFV and suggests a role for complement activation in the pathobiology of infection that will be interesting to follow-up in future studies”.

- Please address the additional minor comments from this reviewer explained in the review.

Reviewer 2.

Reviewer 2 points out that a number of the major concerns raised in the initial review have not been adequately addressed.

1. The authors state that cytokine analyses were performed but were not interpretable even though the controls and standards were valid, hence the exclusion of this data from the manuscript. Without transparent reporting, it is difficult to assess whether negative or contradictory results were selectively excluded.

- Please provide as supplementary information the cytokine analyses you have carried out if feasible.

2. The concern remains that the authors did not directly show that the lectin pathway was responsible for complement activation (line 24). Without directly demonstrating this, the claim that the lectin pathway is responsible is not sufficiently supported. Without proving causality, the findings remain limited and descriptive, and the conclusions are not justified.

- We note that the authors have provided an explanation why additional studies to investigate the role of the lectin pathway in complement activation are unlikely to be conclusive in part due to the current lack of suitable reagents for porcine studies. Please ensure the manuscript makes clear this limitation of your study.

2. T cell responses can be assessed both from fresh and cryopreserved cells and has been used previously in multiple porcine studies (DOI: 10.1128/jvi.00622-24, 10.3389/fimmu.2023.1192604, 10.1038/s41541-025-01212-y). Since this study is complementary to Radulovic et al. 2025 (DOI: 10.3389/fvets.2025.1553310), it is no longer sufficient to leave out the assessment of T cell responses. Measurement of T cell responses would be relevant and necessary to complete this study to support the statement in lines 476-477 that a robust T cell response was induced and what differences in T cell responses could be present in this study in comparison to the work by Lotonin et al. (DOI: 10.7554/eLife.107579.1).

- We note that the authors, in the response to reviewers, agree that T cell responses are important for protection and have added comments and references in the Discussion (Lines 474- 477). The Lotonin et al., article is available as a reviewed preprint but has not yet been revised so a detailed comparison with this preprint may not be useful at this stage of the publication process.

3. The authors have now included the sex of the animals and the concern here is that the sexes were not randomised between the two groups to comply with the ARRIVE guidelines to reduce bias and ensure the robustness of animal studies. Why was there a difference in the sexes of the groups and has this been considered in the analysis and interpretation of these results?

- Please respond to this question.

Please respond to the additional minor points (4 to 18) raised by this reviewer.

Thank you for your responses to the comments from Reviewer 3. We agree these comments have been covered in your revised manuscript and responses to reviewers.

**Journal Requirements:**

At this stage, the following Authors/Authors require contributions: Virginia Friedrichs, Paul Deutschmann, Kerstin Wernike, Tessa Carrau, Martin Beer, Sandra Blome, and Alexander Schäfer. Please ensure that the full contributions of each author are acknowledged in the "Add/Edit/Remove Authors" section of our submission form.

https://journals.plos.org/plospathogens/s/submission-guidelines#loc-parts-of-a-submission

4) We do not publish any copyright or trademark symbols that usually accompany proprietary names, eg ©,  ®, or TM  (e.g. next to drug or reagent names). Therefore please remove all instances of trademark/copyright symbols throughout the text, including:

- ® on pages: 5, and 6.

5) Please upload all main figures as separate Figure files in .tif or .eps format. For more information about how to convert and format your figure files please see our guidelines:

6) We notice that your supplementary Figures are included in the manuscript file. Please remove them and upload them with the file type 'Supporting Information'. Please ensure that each Supporting Information file has a legend listed in the manuscript after the references list.

7) Please amend your detailed Financial Disclosure statement. This is published with the article. It must therefore be completed in full sentences and contain the exact wording you wish to be published.

**Reviewers' Comments:**

Reviewer's Responses to Questions

**Part I - Summary**

Reviewer #1: This manuscript is a new (re)submission of an article that this reviewer has previously reviewed. The authors have addressed the 3 major recommendations and all the minor comments I previously made.

As previously noted, the study provides new and important insight into the duration of immunity against ASFV and suggests a role for complement activation in the pathobiology of infection that will be interesting to follow-up in future studies.

Reviewer #2: The manuscript “Duration of immunity following infection with moderately virulent ASFV” is a resubmission following a previous round of review (PPATHOGENS-D-25-01346) in June this year. The authors have made some limited changes to the manuscript, but a number of the major concerns raised in the initial review have not been adequately addressed. The authors state that cytokine analyses were performed but were not interpretable even though the controls and standards were valid, hence the exclusion of this data from the manuscript. Without transparent reporting, it is difficult to assess whether negative or contradictory results were selectively excluded. Furthermore, the authors have not provided a transparent, point-by-point response to the previous reviews, which makes it difficult to assess what has been revised and how the authors have engaged with the issues raised. Overall, this study is primarily descriptive in nature, relying on broad surface-level profiling to compare responses in animals that survive Armenia08 challenge versus those that do not. While the authors make an initial attempt to explore the involvement of complement, this remains limited to associative observations and does not establish direct causality.

**Part II – Major Issues: Key Experiments Required for Acceptance**

Reviewer #1: None

Reviewer #2: 1. This concern remains that the authors did not directly show that the lectin pathway was responsible for complement activation (line 24). Without directly demonstrating this, the claim that the lectin pathway is responsible is not sufficiently supported. It is regrettable that the authors did not attempt to engage with this critique or mention the limitations of their experimental methods. The scope of PLOS Pathogens states that the journal publishes “original research that clearly demonstrates novelty, importance to its particular field, biological significance, and conclusions that are justified by the study.” This manuscript has not fulfilled all the conditions listed. Without proving causality, the findings remain limited and descriptive, and the conclusions are not justified.

2. T cell responses can be assessed both from fresh and cryopreserved cells and has been used previously in multiple porcine studies (DOI: 10.1128/jvi.00622-24, 10.3389/fimmu.2023.1192604, 10.1038/s41541-025-01212-y). Since this study is complementary to Radulovic et al. 2025 (DOI: 10.3389/fvets.2025.1553310), it is no longer sufficient to leave out the assessment of T cell responses. Measurement of T cell responses would be relevant and necessary to complete this study to support the statement in lines 476-477 that a robust T cell response was induced and what differences in T cell responses could be present in this study in comparison to the work by Lotonin et al. (DOI: 10.7554/eLife.107579.1).

3. The authors have now included the sex of the animals and the concern here is that the sexes were not randomised between the two groups to comply with the ARRIVE guidelines to reduce bias and ensure the robustness of animal studies. Why was there a difference in the sexes of the groups and has this been considered in the analysis and interpretation of these results?

**Part III – Minor Issues: Editorial and Data Presentation Modifications**

Reviewer #1: Minor comments

1. Lines 408-410 and Fig6C: the text “irrespective of viral strain used” does not match the data as the increase in infection rate observed with heat inactivation of the plasma is only significant for macrophages infected with Estonia14 and not those infected with Armenia08. One may speculate that the reduced virulence of the ASFV Estonia14 strain may in part be due to enhanced susceptibility to the lectin pathway of the complement.

2. Legends Fig4 and Fig6A,B: it is not clear which comparisons are indicated by the letters a, b, c, d…

3. Statistics: It is not clear why Fig6C data was analyzed using 1-way ANOVA, when a t test would be sufficient and more appropriate to compare the effects of plasma heat inactivation for each virus strain independently. The comparison of the infection rate between ARM and EST is irrelevant and depends on the experimental conditions.

4. Lines 329 and 207: typo rosette, instead of rousette

Reviewer #2: 4. In line 212, the authors list the ID Screen ASF indirect ELISA which is a multiple antigen ELISA kit for p32, p52 and p72. In section 3.2 and in Fig. 4, the authors then refer to this ELISA and its results and interpret these as responses to the early p32 antigen. How did the authors manage to separate antibody responses to p32 from this multi-antigen ELISA? Furthermore, there is a typo on line 336.

5. I thank the authors for providing the raw data in this resubmission. The author guidelines clearly state that all authors must share the minimal data set which includes the raw data/values of all graphs reported within the manuscript within the supplementary data or in a public data repository. Please can the authors change the Data availability statement to indicate if the raw data will be available in the supplementary data or in a public repository.

6. The authors state within the cover letter that cytokine analyses were performed but were not interpretable even though the controls and standards were valid. How was the interpretability of the results defined? These data would still be relevant as it has been demonstrated by Radulovic and Lotonin (DOI: 10.3389/fvets.2025.1553310, 10.7554/eLife.107579.1). Without transparent reporting, it is difficult to assess whether negative or contradictory results were selectively excluded.

7. In section 2.7, the authors state that data normality was assessed with the D’Agostino-Pearson test. However, the data presented in Figures 6A and B have not fulfilled the assumptions required for a One-way ANOVA. Furthermore, considering that the sera from the same animal is likely to have been treated with the different conditions (PBS, EDTA, EGTA), mixed models would have been more appropriate for this data set, bearing in mind that the data would have to be transformed to normality as the assumptions are not met. A mixed model would also be more appropriate for Fig 6C since the sera from each naïve animal was treated to the four different conditions.

8. Since this resubmission, a detailed systems immunology study investigating T cell responses, cytokine expression and transcriptomics of the study described in Radulovic 2025 (DOI: 10.3389/fvets.2025.1553310) has been submitted to eLife (DOI: 10.7554/eLife.107579.1, July 2025). The authors have missed the opportunity to discuss how this complementary work is relevant to this study, considering that PLOS allows the citation and discussion of relevant preprints.

9. The whole virus lysate ELISA using isotype specific detection antibodies is a modification from the EURL-ASF ELISA, so it should not be referenced as a ‘new’ ELISA, but an adapted or modified ELISA. Please can the authors amend the text accordingly.

10. Line 556: the authors have only demonstrated that the humoral response is induced and maintained in this study. A specific role for antibody responses has not been demonstrated (for example, through depletion studies), hence this phrase should be amended to avoid over-interpretation of the results.

11. Line 147, since four wells were used, this should be quadruplicates and not triplicates.

12. Please can the authors provide information about the calculation method used for determining the virus titres.

13. Is there a reason why statistics were not performed on the data presented in Fig. 1? The authors should include the statistics for Figure 1.

14. For the statistics presented in Figures 4 and 6, what do the different letters represent? Are these all in comparison to the control or to different groups? If these are to different groups, this information should be clearly delineated in the figure legends.

15. Line 81 states that a ‘comprehensive analysis of immune parameters’ was performed. It appears from the manuscript that this is not the case as parameters such as T-cell responses were not included (as outlined by the cover letter). Please can the authors amend this phrase.

16. As mentioned in the previous review, please can the authors provide more background information in the introduction of the manuscript about the choice of complement assays performed to provide the rationale for using these assays. Why did the authors decide to use an ELISA based method over a haemolytic assay?

17. Since the authors have now included the gross pathology scores in the raw data file, these should be referenced in the manuscript such that readers are aware of this data and how it relates to the rest of the study.

18. Line 566-567: please include Figure S3.

PLOS authors have the option to publish the peer review history of their article (what does this mean? ). If published, this will include your full peer review and any attached files.

**Do you want your identity to be public for this peer review?** For information about this choice, including consent withdrawal, please see our Privacy Policy .

Reviewer #1: No

Reviewer #2: No

**Figure resubmission:**
---

## [Decision Letter · Decision Letter 1]

17 Feb 2026

PPATHOGENS-D-25-02903R1

Duration of immunity following infection with moderately virulent ASFV

PLOS Pathogens

Dear Dr. Blome,

Thank you for submitting your manuscript to PLOS Pathogens. After careful consideration, we feel that it has merit but does not fully meet PLOS Pathogens's publication criteria as it currently stands. Therefore, we invite you to submit a revised version of the manuscript that addresses the points raised during the review process.

We look forward to receiving your revised manuscript.

Kind regards,

Linda Kathleen Dixon

Academic Editor

PLOS Pathogens

Donna Neumann

Section Editor

PLOS Pathogens

Sumita Bhaduri-McIntosh

Editor-in-Chief

PLOS Pathogens

orcid.org/0000-0003-2946-9497

Michael Malim

Editor-in-Chief

PLOS Pathogens

orcid.org/0000-0002-7699-2064

**Additional Editor Comments:**

Thank you for responding to the reviewer and editorial comments. The minor revisions requested by the reviewer should be addressed.

**Reviewers' Comments:**

Reviewer's Responses to Questions

**Part I - Summary**

Reviewer #2: I thank the authors for engaging with the review, providing the Luminex data and for improving the manuscript.

**Part II – Major Issues: Key Experiments Required for Acceptance**

Reviewer #2: (No Response)

**Part III – Minor Issues: Editorial and Data Presentation Modifications**

Reviewer #2: I have the following minor comments:

1.I thank the authors for addressing the question regarding randomisation of sexes. Could the authors please add their discussion on the absence of sex randomisation in the Discussion section of the manuscript?

2.Line 159-160: If I have understood the response correctly, the authors have clarified that the titrations were performed on cells from three different pigs? In that case the phrasing is confusing in the manuscript. The four wells would be the technical replicates and the three repeats in three different animals would be biological replicates. In that case, could the authors please amend the sentence to improve clarity and readability?

PLOS authors have the option to publish the peer review history of their article (what does this mean? ). If published, this will include your full peer review and any attached files.

**Do you want your identity to be public for this peer review?** For information about this choice, including consent withdrawal, please see our Privacy Policy .

Reviewer #2: No

**Figure resubmission:**
---

## [Editor Report · Decision Letter 2]

19 Feb 2026

Dear Dr. Blome,

We are pleased to inform you that your manuscript 'Duration of immunity following infection with moderately virulent ASFV' has been provisionally accepted for publication in PLOS Pathogens.

Best regards,

Linda Kathleen Dixon

Academic Editor

PLOS Pathogens

Donna Neumann

Section Editor

PLOS Pathogens

Sumita Bhaduri-McIntosh

Editor-in-Chief

PLOS Pathogens

orcid.org/0000-0003-2946-9497

Michael Malim

Editor-in-Chief

PLOS Pathogens

orcid.org/0000-0002-7699-2064
---

## [Editor Report · Acceptance letter]

Dear Dr. Blome,

We are delighted to inform you that your manuscript, "Duration of immunity following infection with moderately virulent ASFV," has been formally accepted for publication in PLOS Pathogens.

Best regards,

Sumita Bhaduri-McIntosh

Editor-in-Chief

PLOS Pathogens

orcid.org/0000-0003-2946-9497

Michael Malim

Editor-in-Chief

PLOS Pathogens

orcid.org/0000-0002-7699-2064